# Dietary Curcumin Prevented Astrocytosis, Microgliosis, and Apoptosis Caused by Acute and Chronic Exposure to Ozone

**DOI:** 10.3390/molecules24152839

**Published:** 2019-08-05

**Authors:** Sendar Daniel Nery-Flores, Mario Alberto Ramírez-Herrera, María Luisa Mendoza-Magaña, Marina María de Jesús Romero-Prado, José de Jesús Ramírez-Vázquez, Jacinto Bañuelos-Pineda, Hugo Alejandro Espinoza-Gutiérrez, Abraham Alberto Ramírez-Mendoza, Mariana Chávez Tostado

**Affiliations:** 1Laboratorio de Neurofisiología, Departamento de Fisiología, Centro Universitario de Ciencias de la Salud, Universidad de Guadalajara, 44340 Guadalajara, Jalisco, México; 2Departamento de Medicina Veterinaria, Centro Universitario de Ciencias Biológicas y Agropecuarias, Universidad de Guadalajara, 45200 Zapopan, Jalisco, México; 3Departamento de Clínicas de la Reproducción Humana, Crecimiento y Desarrollo Infantil, Centro Universitario de Ciencias de la Salud, Universidad de Guadalajara, 44340 Guadalajara, Jalisco, México

**Keywords:** curcumin, ozone, astrocyte, microglia, apoptosis, hippocampus

## Abstract

Ozone is the most oxidant tropospheric pollutant gas, causing damage through the formation of reactive oxygen and nitrogen species. Reactive species induce the nuclear factor-kappa B (NF-κB) activation leading to neuroinflammation characterized by astrocytosis, microgliosis, and apoptotic cell death. There is interest in evaluating the pharmacological activity of natural antioxidants to confer neuroprotection against the damage caused by ozone in highly polluted cities. Curcumin has been proven to exert a protective action in the central nervous system (CNS) of diverse experimental models, with no side effects. The aim of this work is to evaluate the effect of curcumin in a preventive and therapeutic manner against the astrocytosis, microgliosis, and apoptosis induced by ozone in rat hippocampus. Fifty Wistar rats were distributed into five experimental groups: The intact control, curcumin fed control, ozone-exposed group, and the preventive and therapeutic groups receiving the curcumin supplementation while exposed to ozone. Ozone caused astrocytosis and microgliosis, as well as apoptosis in the hippocampus. Meanwhile, curcumin was able to decrease the activation of microglia and astrocytes, and apoptotic cell death in both periods of exposure. Therefore, we propose that curcumin could be used as a molecule capable of counteracting the damage caused by ozone in the CNS.

## 1. Introduction

Air pollution affects the health of millions of people around the world [1]. Ozone (O_3_) is a hazardous tropospheric pollutant that causes oxidative damage, and the symptoms associated with its exposure have been mainly documented in respiratory tract diseases. However, the adverse effects of O_3_ have also been described in the central nervous system (CNS) [2,3]. The interaction of O_3_ with the alveolar epithelium leads to the formation of reactive oxygen and nitrogen species (RONS) that diffuse into the bloodstream and damage biomolecules of the whole body [4,5]. Also, RONS reach brain structures through the olfactory pathway causing oxidative damage, neuroinflammation, and apoptosis [3,4,6].

Due to the high oxygen and energy demand of the CNS, oxidative damage and inflammation are easily established. Furthermore, nerve tissue contains high levels of transition metals that catalyze the formation of the reactive hydroxyl radical [7,8]. The hippocampus is markedly more sensitive than other CNS structures like the brain cortex, cerebellum, or striatum to oxidative damage caused by O_3_. Thus, its study is relevant due to its fundamental role in cognition, memory, and mood control [9,10]. Within the hippocampus, there is a differential regional susceptibility due to the level of expression of antioxidant enzymes like superoxide dismutase and glutathione peroxidase [11]. The sensibility of the hippocampus to oxidative damage is also increased by a strong cell packing in the Cornu Ammonis (CA) and dentate gyrus [12]. Other brain structures like striatum and substantia nigra are more sensitive to oxidative damage caused by other toxic agents like tributyline [13].

Astrocytes, as well as microglial cells, are involved in the inflammatory response and undergo reactive modifications revealed by changes in their structural, metabolic, and gene expression profile [14,15]. Astrocytes maintain the homeostasis of the CNS through the conformation of the brain blood barrier together with the endothelial system, and regulate the availability of nutrients, synthesis of neuromodulators, and neurotransmitter precursors, which lead to a profound modulation of neuronal function [16]. Glial fibrillary acidic protein (GFAP), a kind of intermediate microfilament whose expression level (mRNA and protein) is extensively used to identify astrocytes and their reactivity level, is strongly associated to physiological and pathological conditions [17]. An increase in the expression of GFAP is related to the activation of astrocytes. Furthermore, high levels of GFAP have been associated with neurodegenerative diseases, mainly Alzheimer’s disease [18].

Under an altered condition in the neuronal microenvironment, profusely ramified microglia cells detect changes and initiate the innate immune response that includes phagocytosis, antigen presentation, releasing cytokines, neurotrophic factors, and inflammation mediators, as well as overproduction of RONS [19,20,21]. Exogenous RONS act as activators of NF-κB, which in turn induces the synthesis and release of inflammatory cytokines and pro-oxidant enzymes [22]. In addition, microglia cells undergo important phenotype changes that include the retraction of their cytoplasmic prolongations acquiring a phagocytic phenotype (M1) to eliminate injured cells [23]. After the injury has been controlled, microglia adopts a phenotype (M2) that promotes nerve tissue regeneration [21,22].

The persistent reactive gliosis caused by O_3_ and RONS also leads to mitochondrial dysfunction with energy failure, which in turn, leads to proapoptotic events that finally cause cell death [6]. Apoptosis occurs in neurons, astrocytes, oligodendrocytes, and even in microglial cells. Neuronal apoptosis is involved in a variety of pathological conditions including epilepsy, Alzheimer’s, and Parkinson’s disease, multiple sclerosis, excitotoxicity, and senile dementia, among others [24].

In spite of the hazardous effects of the oxidative damage caused by O_3_ in the CNS, few attempts have been done to diminish its impact on human health. Some synthetic and natural molecules have been tested for their antioxidant activity against O_3_ toxicity as an air pollutant [25,26,27]. However, there is major concern on the adverse side effects of these compounds tested to date.

In our previous work, we reported that curcumin (CUR) exerted an antioxidant and anti-inflammatory effect in a preventive and therapeutic approach against the damage caused by acute and chronic exposure to O_3_ in a model of oxidative stress [28]. Moreover, the activation of NF-κB was efficiently inhibited by CUR, which would lead to the significant decrease of inflammatory interleukins. This molecule is a natural phenolic compound isolated from the rhizome of the Asian plant *Curcuma longa* Linn [29]. The pharmacological activities documented for CUR include antioxidant, anti-inflammatory, antiapoptotic, and neuroprotective properties [30,31,32]. Furthermore, it has been proven that CUR exerts no adverse effects, even at high doses during long-term administration (up to 8 g/day for three months) [33]. The antioxidant effects of CUR are evident upon mitochondrial function by neutralizing reactive species generated within the mitochondria, as well as increasing the expression of antioxidant enzymes controlled through the activation of the nuclear factor erythroid 2-related factor 2 (Nrf2) [34,35,36,37].

This investigation was envisioned to elucidate the regulatory effect of dietary CUR in the inflammatory and apoptotic cellular response of astrocytes and microglia in the rat hippocampus under O_3_ exposure during the acute and chronic periods.

## 2. Results

### 2.1. Curcumin Reduced Activation of Astrocytes in the Rat Hippocampus

The activation of astrocytes was reflected by an increase in the immunoreactivity level of GFAP. The acute response of the CA1 and CA3 regions in the rat hippocampus under oxidative stress was shown with a significant increase in the number of astrocytes in the acute ozone control (AOC) group (*p* < 0.001), compared with the CA1 and CA3 regions of the acute intact control (AIC) group and acute curcumin control (ACC) group. Both, the acute preventive (AP) and acute therapeutic (AT) groups presented a significant decrease in astrocyte activation compared to the AOC group (CA1, *p* < 0.001 and CA3, *p* < 0.001). In addition, the AP group presented a significant decrease in comparison with the AT group in the CA1 (*p* < 0.001) and CA3 (*p* < 0.01) regions (Figure 1).

In the chronic phase, the chronic ozone control (COC) group showed a significant increase in the number of activated astrocytes (CA1 and CA3, *p* < 0.001) compared to the chronic intact control (CIC) and the chronic curcumin control (CCC) group. The chronic preventive (CP) group and the chronic therapeutic (CT) group showed a significant reduction in astrocyte activation with respect to the COC group (*p* < 0.001). Finally, both groups CP and CT did not show significant differences between them in both hippocampal regions (Figure 2).

### 2.2. Curcumin Decreased the Microgliosis in the Rat Hippocampus

The activation of the microglia was determined according to the change of morphology, going from the resting state (RS) to the activated state (AS). The exposure to O_3_ caused a significant increase in the number of microglia in the AS (AOC group, CA1 and CA3, *p* < 0.001) compared to the AIC and ACC groups, as well as a significant decrease in the number of RS microglial cells compared to the control groups (CA1 and CA3, *p* < 0.001). The effect of preventive CUR administration in the AP group caused a significant decrease in the number of microglia in the AS (CA1 and CA3, *p* < 0.001) compared to the AOC group. Conversely, this group showed a greater number of microglia in the RS (CA1, *p* < 0.01 and CA3, *p* < 0.05). This same phenomenon was observed in the AT group with a significant decrease in the number of microglia in the AS (CA1 and CA3, *p* < 0.001) and consequently, an increase in the number of microglia in the RS (CA1, *p* < 0.001; CA3, *p* < 0.01). It is important to note that the AP group showed a significant decrease in activated microglia with respect to the AT group (*p* < 0.01) in the CA3 region (Figure 3).

In the chronic phase, the COC group showed a significant increase in the number of microglia in the AS (CA1 and CA3, *p* < 0.001); in addition, a similar number of cells in the RS were maintained in CA1 and CA3 regions. The CP group showed a significant decrease of microglia in the AS (CA1, *p* < 0.01; CA3, *p* < 0.05) compared to the COC group and the number of cells in the RS in CA1 and CA3 was similar to the COC group. The CT group presented a significant decrease of microglial cells in the AS (CA1, *p* < 0.001; CA3, *p* < 0.01), and a similar number of cells in the RS in CA1 and CA3 regions with respect to the COC group. The CT and CP groups did not show significant differences between them (Figure 4).

In addition, a significant increase was observed in the total number of microglial cells in the group exposed to ozone during the acute phase in the CA3 region of the hippocampus. The CA1 region showed no significant difference between the groups (Figure 5A). This effect was more noticeable during the chronic phase of exposure, where in both approaches, CUR significantly reduced the total number of microglia in both regions of the hippocampus, however, the count remained increased compared to the intact group (Figure 5B).

### 2.3. Curcumin Decreased Apoptosis Levels in Rat Hippocampus

The deleterious effect of O_3_ in the CNS was now evidenced by its final apoptotic effect. In turn, CUR prevented and decreased cell death in the CA1 and CA3 regions of the rat hippocampus. The acute exposure to O_3_ caused a significant increase in the percentage of hippocampal apoptotic cells (CA1 and CA3, *p* < 0.001) compared to the AIC and ACC groups. The preventive antiapoptotic effect of CUR caused a significant decrease in CA1 and CA3 regions of the hippocampus (CA1, *p* < 0.01 and CA3, *p* < 0.001). The therapeutic effect of CUR caused a significant decrease in the apoptotic percentage of CA1 and CA3 cells (CA1, *p* < 0.05; CA3, *p* < 0.001) compared to the corresponding regions of the AOC group. The AT and AP groups did not show significant differences between them (Figure 6).

The chronic exposure to O_3_ caused a higher percentage of apoptotic cells than in the acute exposure (*p* < 0.001). The COC group showed a significant increase in the percentage of apoptotic cells (CA1 and CA3, *p* < 0.001). The neuroprotective effect of CUR caused a significant decrease of apoptotic cells in both, the preventive (CA1, *p* < 0.05; CA3, *p* < 0.01) and the therapeutic approaches (CA1, *p* < 0.05; CA3, *p* < 0.01), compared to the COC group (Figure 7).

## 3. Discussion

In this work, we report that CUR, administrated in the preventive and therapeutic manners, exerted a neuroprotective effect against the damage caused by O_3_ during acute and chronic exposure in the hippocampus of the rat. We demonstrated that CUR decreased astrocytosis, microgliosis, and apoptosis in the CA1 and CA3 regions from rat hippocampus.

Based on a well-known model of O_3_-induced oxidative stress, we demonstrated in a previous work that the acute and chronic exposure to O_3_ caused neuronal tissue lipid peroxidation and protein oxidation, as well as serum levels increase of interleukine-1β, (IL-1β), interleukine-6 (IL-6), and tumoral necrosis factor-α, (TNF-α). The administration of CUR decreased all these parameters in the preventive and therapeutic approaches. Additionally, we showed that these approaches were also efficient in reducing the activation level of NF-κB in the hippocampus of rats after acute or chronic exposure to O_3_ [28].

Air pollutants cause oxidative stress as a fundamental process that leads to an inflammatory response [38]. Exposure to O_3_ has been associated with respiratory and cardiovascular problems, and more recently, with structural and functional alterations of the CNS [4,6,39]. In this model of oxidative stress caused by O_3_, we showed that this gas is capable of causing astrocytosis, microgliosis, and neuronal apoptosis in the hippocampus, in accordance with other studies [4,6].

Astrocytes are the most abundant population of glial cells in the CNS and they play an essential role in neuronal homeostasis. Astrocytes respond to CNS damage with hypertrophy and cell proliferation, a process known as “reactive astrocytosis” [16,40]. Our results show that the acute exposure to O_3_ caused an increased immunoreactivity to GFAP, as well as an increase in the number of astrocytes in the CA1 and CA3 regions of the hippocampus. This same effect was maintained after a chronic exposure for 60 days. This is consistent with other investigations where exposure to O_3_ caused an increased number of astrocytes in the hippocampus from 15 to 90 days [4]. This increase in the astrocyte cell population is also related to a higher expression of GFAP, and is concomitant with the increased activation of NF-κB [28,41]. Such activation is essential for the synthesis of GFAP and inflammatory mediators released by astrocytes such as IL-1β, IL-6, and TNF-α [42,43].

The preventive and therapeutic administrations of CUR were able to significantly reduce the astrocytosis during the acute and chronic exposure phases. In addition, the beneficial effect of CUR was more evident during the acute phase of exposure compared to the chronic phase. In the acute phase, the administration of CUR in a preventive manner presented a greater effect than in the therapeutic mode, which suggests that the antioxidant effect of CUR prevents the activation of astrocytes by RONS generated during the exposure to O_3_ [44]. Moreover, CUR was able to modulate the activity of astrocytes through its antioxidant and anti-inflammatory effects and CUR reduced the expression of GFAP and inflammatory cytokines by inhibiting the activation of NF-κB [45,46,47,48]. However, when the O_3_ exposure produced damage before CUR administration, we achieved a lower reduction in the acute astrocyte activation. Meanwhile, in the chronic exposure to O_3_, the preventive and therapeutic approaches seemed to act with similar efficacy.

Microglia plays an important role in neuronal homeostasis and regulation of the innate immune response [49]. Microglial cells in the resting state exhibit a branched phenotype, allowing the search for alterations in the extracellular environment or the presence of cellular debris [23]. If an oxidative damage process occurs, i.e., induced by ozone exposure, the microglia changes to an activated state, acquiring an amoeboid appearance and releasing proinflammatory mediators as cytokines, pro-oxidant enzymes (inducible nitric oxide synthase, cyclo-oxigenase-2, 5-lipoxigenase), and growth factors, among others [50].

In our experiments, the preventive and therapeutic administration of CUR showed an excellent modulating effect on microglia activation after acute and chronic exposure to O_3_. CUR increased the number of microglia at the resting state compared to cells found in the activated state, which suggests an anti-inflammatory effect. The neuroprotective effect of CUR was more evident in the acute phase with a greater decrease in the number of activated microglia than the chronic phase. Furthermore, the preventive administration of CUR in the acute exposure phase to O_3_ clearly reduced the activation of the microglia, especially in the CA3 region. Other studies have demonstrated that a single exposure to 1 ppm of O_3_ for 4 h is not capable of releasing classic inflammatory cytokines (TNF-α, IL-1β, IL-6) in the lungs, however, microglia cells became activated, thus RONS could be reaching the CNS even during a short-term exposure [51]. The change of phenotype of the microglia is due to the increase in RONS production, causing the activation of kinases that stimulate the phosphorylation of IκB and consequently, the activation of NF-κB occurs [41,52]. Activation of NF-κB is a critical event that promotes the shift of microglia towards the activated phenotype [53]. Several reports have demonstrated that CUR inhibits NF-κB activation, and consequently, the phenotype change of the microglia [54,55,56,57]. In our experiments, CUR was capable of modulating microglia both in the preventive and therapeutic administration. In the preventive approach, the resting phenotype was preserved despite the oxidative damage caused by O_3_. This could be due to an anticipated action of CUR as an inhibitor of the NF-κB pathway, inhibiting the activation of IKK. Furthermore, the preventive administration of CUR was able to activate Nrf2 leading to the increased expression of antioxidant enzymes, as well as inactivation of apoptosis induced by p53 [55,58]. In contrast, the therapeutic administration showed a lower effect once the microglia had been previously activated by O_3_ acquiring the activated phenotype and NF-κB activation had been previously established. Thus, CUR did not completely revert this activation. A plausible explanation could be that CUR was able to stimulate the shift to an anti-inflammatory cytokine profile by activating the Nrf2 pathway, however, this remains to be elucidated in future experiments that could determine the distribution of microglial population in the three different functional states [57,59].

The repeated exposure to O_3_ produces a state of oxidative stress capable of causing cell death [6]. In the present work, the model used to cause ozone-induced oxidation has been demonstrated to be time-dependent. Accordingly, the exposure to O_3_ elapses, oxidative damage, astrocytosis, and microgliosis are perpetuated, and as a result, apoptosis increases. The cellular intrinsic pathway as well as the stress induced in endoplasmic reticulum would contribute to an increase in the expression of Bax and caspase-3, which trigger the programmed cell death process [60]. In the acute and chronic exposure to O_3_, the most susceptible region to the apoptotic process was CA3. However, it has been documented that CA1 neurons are more vulnerable than CA3 neurons in experimental cerebral ischemia, as well as in the early stages of Alzheimer’s disease, aging, and oxidative stress [61]. The high susceptibility of CA1 neurons is due to the presence of RONS in this region, even under normal conditions. When a stimulus generates an exceeded production of RONS, the antioxidant response in CA1 may become depleted and lead to cell death [11].

CUR proved to have an important antiapoptotic effect against the damage caused by O_3_, expressly in the preventive administration during the acute phase. In the chronic phase, CUR exhibited its antiapoptotic effect in both the preventive and therapeutic approaches. Our findings suggest that the antiapoptotic effect of CUR was more evident during the acute process of damage and its activity was less effective in the chronic phase. According to our results, the persistent exposure to O_3_ could possibly require a dynamic adjustment of the initial dose of CUR and thus, improve the efficiency of the neuroprotective effect. The antiapoptotic mechanisms of CUR are the reduction of cytochrome C translocation, the decrease in Bax expression, the increase in Bcl-2 level, and the inhibition of caspases 3 and 9 activity [62,63,64,65]. In addition, the preventive approach presented a more pronounced antiapoptotic effect, possibly due to an anticipated increase in Bcl-2 levels and through the enhanced activity of major antioxidant enzymes like superoxide dismutase, catalase, glutathione reductase, and glutathione peroxidase through the activation of Nrf2 by CUR [66]. Therefore, CUR is a molecule capable of protecting neurons from oxidative damage and inflammation caused by different agents or processes and consequently, decreasing apoptotic cell death. Moreover, it has been shown that CUR exerts a significant antiapoptotic effect in different models of acute or chronic neuronal damage using diverse administration routes, formulations, and different doses [66,67,68,69,70,71].

The only natural molecules that have been studied against air pollutants other than ozone are vitamin C and E, but such studies do not analyze the damage to the CNS, however the mechanisms of action are mainly as ROS scavengers. Synthetic molecules like tibolone [72] and taurine [25] have been previously reported against damage caused by ozone in CNS, but the risk of side effects limits the administration of these molecules. Also, the protective effect of capsaicin against inflammation of airways of guinea pigs caused by ozone has been reported [73].

After analyzing diverse scientific reports, it seems plausible to propose the execution of studies conducted on human subjects living in highly polluted cities with an intermittent high concentration of O_3_ and the impact of CUR in physical, mental health, and cognitive performance.

## 4. Materials and Methods

### 4.1. Animals

This study was conducted with 50 male Wistar rats (*Rattus norvegicus*), 21 days old, weighing ≈ 130 g which were kept under light/dark cycles 12 × 12 h, 22 ± 2 °C and relative humidity of 50–60% with free access to water and food. Experimentation was done according to the National Institutes of Health guide for the care and use of laboratory animals (NIH Publications No. 8023, revised 1978) which are established in the Ethical Committee of the Health Science Center (CUCS, Universidad de Guadalajara).

### 4.2. Diet

CUR was incorporated into food pellets using an alcoholic extract. After heating the pellets at 60 °C for 4 h, ethanol was eliminated and the concentration of CUR was determined by UV-spectrophotometry at λ 230 after ethanolic extraction from samples of food pellets [28,71]. The turmeric alcoholic extract was prepared from commercial curcuma (AMBE Phytoextracts PVT LTD, New Delhi, India, Kosher, Batch No. 09076), the concentration of CUR was determined by UV spectrophotometry, and the molecular identity was analyzed by infrared spectrometry and compared with a CUR standard (Sigma Chemical Co., St. Louis, MO, USA). The daily dose of CUR to animals was approximately 5.6 mg/kg body weight.

### 4.3. Experimental Design

Five experimental groups were randomly organized with ten rats each. A seven day adaptation phase was applied before the beginning of the experiments to minimize the effect of human contact, food, and the lodging place in the experimental model. Two phases of experimentation were established: An acute phase (A, 15 days) and a chronic one (C, 60 days). Also, the modes of CUR supplementation in the experimental groups were defined as preventive (P) or therapeutic (T). Thus, each group was subdivided in 2 identifying the following conditions: The O_3_ control groups that were exposed to 0.7 ppm of O_3_ during the acute and chronic phase (AOC and COC, *n* = 5); the acute and the chronic control intact groups (AIC and CIC, *n* = 5) exposed to O_3_–free air, without CUR; the CUR controls groups that received the CUR supplementation, without O_3_ exposure in the same time periods (ACC, *n* = 5, and CCC, *n* = 5); for the therapeutic mode, rats were exposed to 0.7 ppm of O_3_ during seven days followed by dietary administration of CUR until the end of both exposure times (AT and CT, *n* = 5). The preventive mode (AP, *n* = 5 and CP, *n* = 5) received food supplementation with CUR for 7 days before exposure to O_3_.

### 4.4. Ozone Exposure

The daily dose of O_3_ was 0.7 ppm for 4 h, and the acute exposure phase was 15 days and 60 days for the chronic phase. A hermetic acrylic chamber (65 × 25 × 45 cm L/H/D), was connected to a gas premix chamber (40 × 24 × 45 cm) that received the O_3_ generated by a Certizon C100 ozonizer (Sander, Elektroapparatebau GmbH, Uetze, Germany) connected to a medical grade oxygen tank. To adjust the required concentration, the O_3_ was mixed with O_3_–free air, and was monitored with a semiconductor sensor (ES-600, Ozone Solutions Inc., Hull, IA, USA) with a constant flow of 1.6–1.2 L/min. The contained O_3_ was expelled from the chamber and inactivated with a neutralizing filter.

### 4.5. Tissue Samples

The rats of each group (*n* = 5) were anesthetized with sodium pentobarbital (40 mg/Kg) and intracardially perfused with PBS with 1200 U of heparin, followed by perfusion with 4% paraformaldehyde (Sigma Chemical Co., St. Louis, MO, USA) in PBS for 15 min. The brain was removed and postfixed by immersion in 4% paraformaldehyde for 48 h. Then, a tissue slice was obtained by two coronal cuts at the stereotactic coordinates –6.04 and –2.80 from Bregma and tissue slices were placed in a cryoprotective solution (30% saccharose, 30% ethylene glycol, PBS pH 7.2). Brain tissue samples containing the hippocampus were cut at 18 µm on a cryostat (Leica CM1850, Nussloch, Baden-Württemberg, Germany) and mounted on gelatin coated coverslips.

### 4.6. Immunohistochemistry for Astrocytes and Microglia

Astrocyte immunoreactivity was detected by the use of a chicken polyclonal antibody against GFAP (AB5541 Millipore, Burlington, MA, USA) diluted 1:200 and incubated for 18 h at 4 °C. A peroxidase-labeled rabbit anti-chicken IgG (ab97140 Abcam, Cambridge, MA, USA) diluted 1:250 was used to incubate tissue sections for 1.5 h at room temperature. The reaction product was visualized with a solution of 0.01 M Tris-HCl pH 7.6, 0.06% of DAB, 0.003% MgCl_2_, and 0.03% H_2_O_2_ for 5 min. The reaction was stopped with distilled water. Counterstaining was performed with hematoxylin for 30 s, dehydrated, and mounted with synthetic resin. The slides were examined with an Olympus IX-71 Inverted Microscope and photographed with a CoolSnap-Pro Color Digital Camera (Roper Scientific, Tucson, AZ, USA). The total number of positive cells was counted per field (200×) in three different fields in CA1 and CA3.

The activated microglia were visualized with isolectin-B4 coupled to Alexa FluorTM 488 (I21411, Invitrogen, Burlington, Ontario, Canada) dissolved in PBS (1:50). The tissue sections were incubated for 24 h at 4 °C and then, washed with PBS and mounted in buffered glycerol. The slides were examined with an Olympus IX-71 Inverted Microscope and photographed with a CoolSnap-Pro Color Digital Camera. The total positive cells were counted per field (100×), in three different fields and classified as: Rest (RS) or activated (AS) state according to their morphology, considering cells at rest as branched phenotype and active cells as amoeboid morphology.

### 4.7. Fragment End Labeling of DNA (FragEL)

Apoptotic cells were identified by TdT-FragEL DNA Fragmentation Detection Kit (Cat #QIA33, Calbiochem Biosciences, La Jolla, CA, USA). CA1 and CA3 hippocampus histological sections were processed as follows: The sections were rinsed with Tris-buffered saline 0.02 M pH 7.6 (TBS) and incubated with proteinase K (20 μg/mL) for 10 min and endogenous peroxidase was blocked. Afterwards, they were incubated with terminal deoxynucleotidyl transferase enzyme (TdT), which catalyzes the addition of biotin-labeled and unlabeled deoxynucleotides. Biotinylated nucleotides were detected using a streptavidin-peroxidase conjugate and visualized with DAB-urea solution with metal enhancer. Sections were counterstained with 0.1% methyl green and mounted with synthetic resin. The slides were examined with an Olympus IX-71 Inverted Microscope and photographed with a CoolSnap-Pro Color Digital Camera. The percentage of apoptotic cells was obtained by counting the number of positive cells from a total of one hundred cells in three different fields (200×). The CA1 and CA3 regions of the hippocampus were evaluated.

### 4.8. Statistical Analysis

The data are expressed as the mean ± standard error of the mean (SEM). Data were performed using nonparametric Kruskal–Wallis and Mann–Whitney U-tests. In all experiments, *p* < 0.05 were considered statistically significant. GraphPad Prism 6.01 software (GraphPad Software Inc., La Jolla, CA, USA) was employed for all analyses.

## 5. Conclusions

The results of this work demonstrate that the administration of CUR decreases the astrocytosis, microgliosis, and apoptosis in the hippocampus induced by exposure to O_3_. CUR could be a safe and efficient alternative to prevent CNS damage caused by exposure to O_3_.

## Figures and Tables

**Figure 1 molecules-24-02839-f001:**
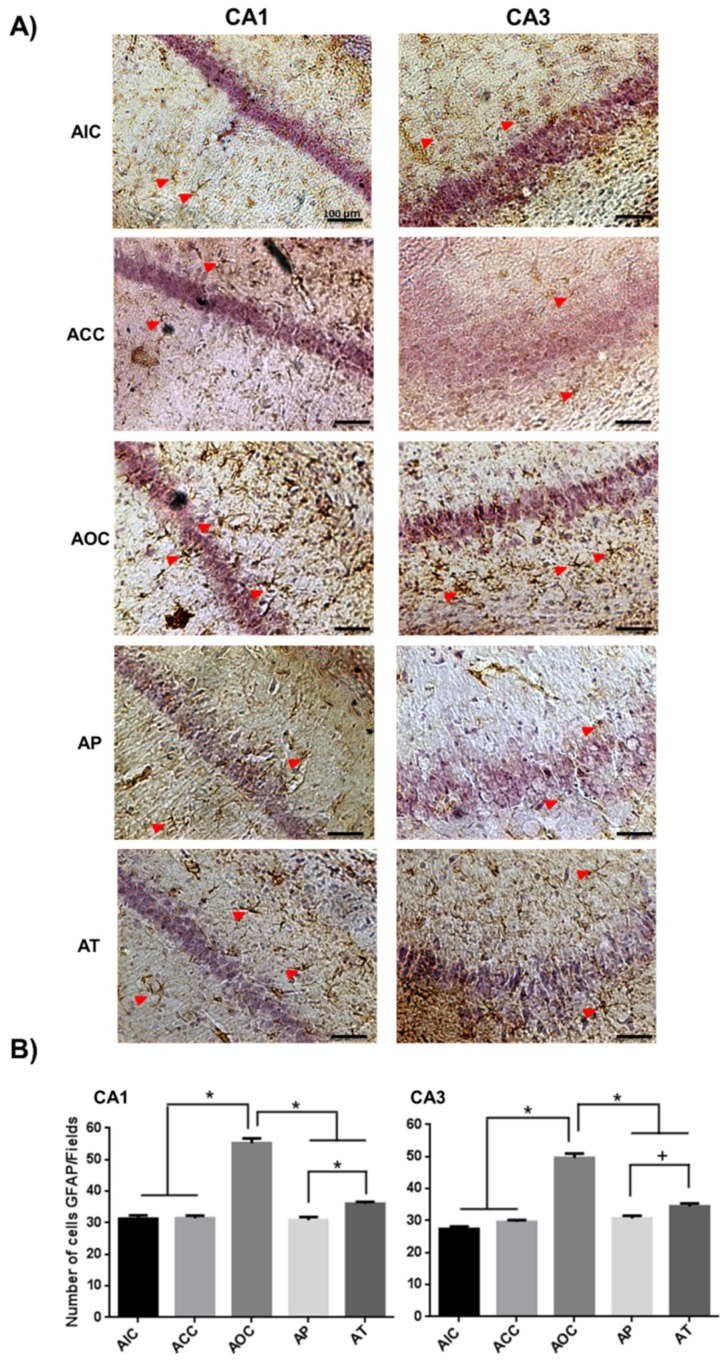
Effect of curcumin (CUR) on the astrocyte activation caused by acute exposure to O_3_. (**A**) Representative light photomicrographs show glial fibrillary acidic protein (GFAP) immunoreactivity (red arrowhead) in CA1 and CA3 regions of the rat hippocampus in the acute phase. The acute preventive (AP) and acute therapeutic (AT) groups showed decreased cell immunoreactivity. Scale bars 100 μm. (**B**) GFAP immunoreactive cells: The quantitation of GFAP positive cells from experimental groups was depicted in this graph as the mean ± SEM. * *p* < 0.001, + *p* < 0.01.

**Figure 2 molecules-24-02839-f002:**
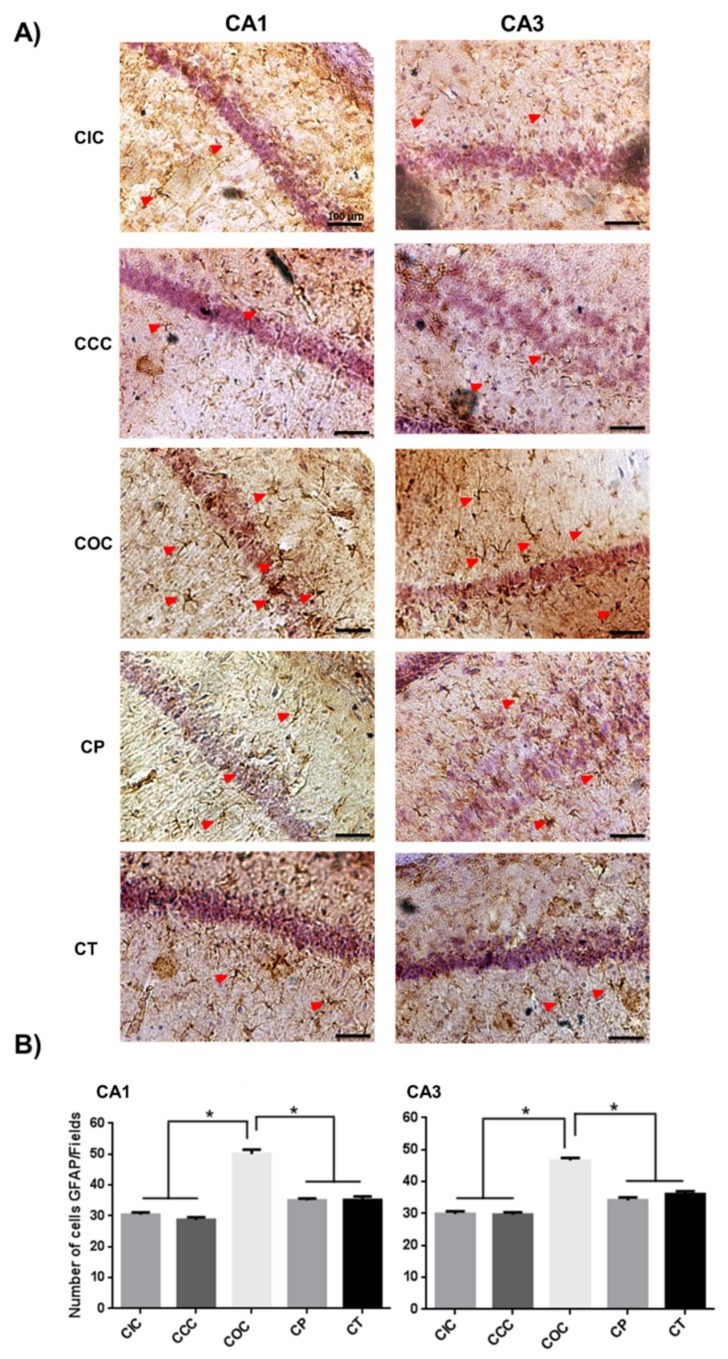
Effect of CUR on the astrocyte activation caused by chronic exposure to O_3_. (**A**) Representative light photomicrographs show GFAP immunoreactivity (red arrowheads) in CA1 and CA3 regions of the rat hippocampus in the chronic phase. The chronic ozone control (COC) group presents a larger number of GFAP immunoreactive cells than the chronic intact control (CIC) group. The chronic preventative (CP) and chronic therapeutic (CT) groups showed a significant decrease in cell immunoreactivity. Scale bars 100 μm. (**B**) Total quantitation of immunoreactive cells are shown in the graph as mean ± SEM. * *p* < 0.001.

**Figure 3 molecules-24-02839-f003:**
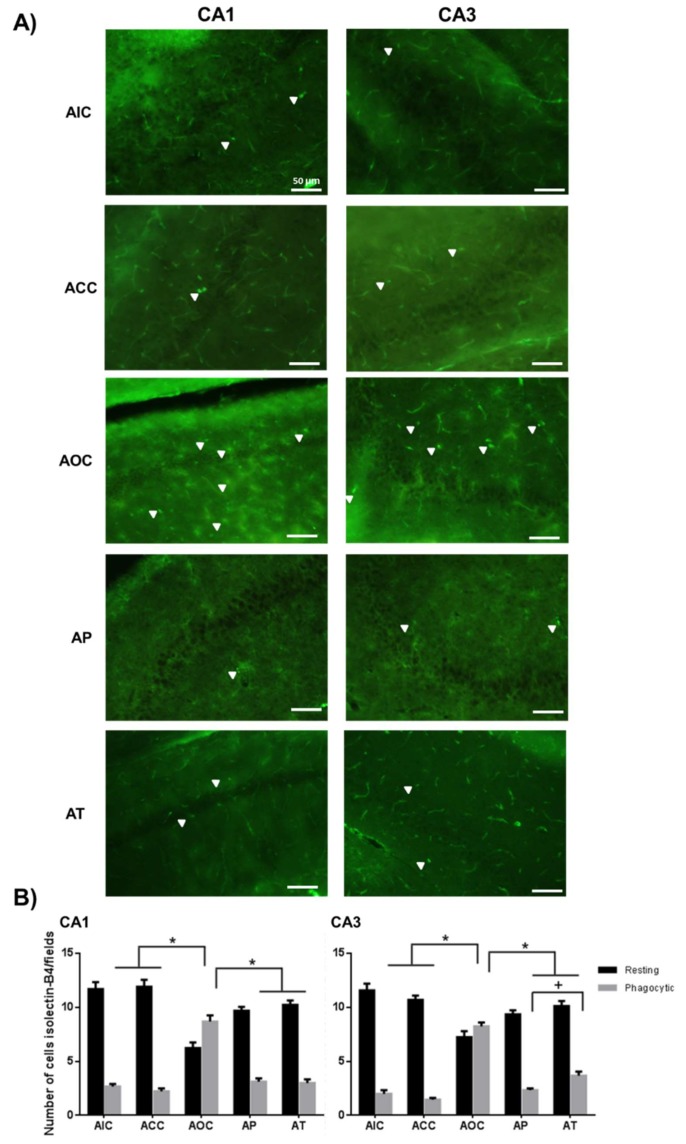
Effect of CUR on microglial cells in the acute exposure to O_3_. (**A**) Representative fluorescent photomicrographs show isolectin-B4 reactivity in the CA1 and CA3 regions of the rat hippocampus. The AP and the AT groups showed a significant decrease in the number of activated microglia. Scale bars 50 μm. (**B**) The total number of isolectin-B4 positive cells are shown in the graph as mean ± SEM and classified according to their morphology. * *p* < 0.001, + *p* < 0.01.

**Figure 4 molecules-24-02839-f004:**
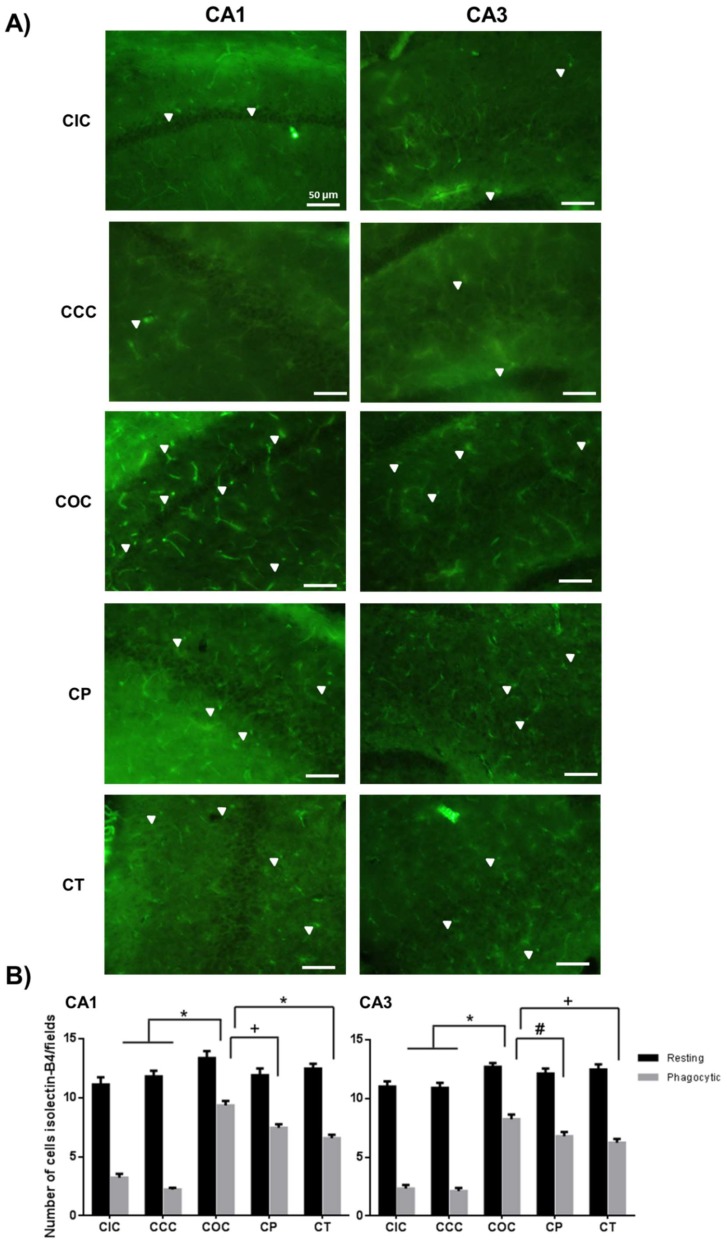
Effect of CUR on microglial cell activation in the chronic exposure to O_3_. (**A**) Representative fluorescent photomicrographs show isolectin-B4 reactivity in the CA1 and CA3 regions of the rat hippocampus. The CP and the CT groups showed a significant decrease in the number of activated microglia. Scale bars 50 μm. (**B**) The total number of isolectin-B4 positive cells are shown in the graph as mean ± SEM and classified according to their morphology. * *p* < 0.001, + *p* < 0.01, # *p* < 0.05.

**Figure 5 molecules-24-02839-f005:**
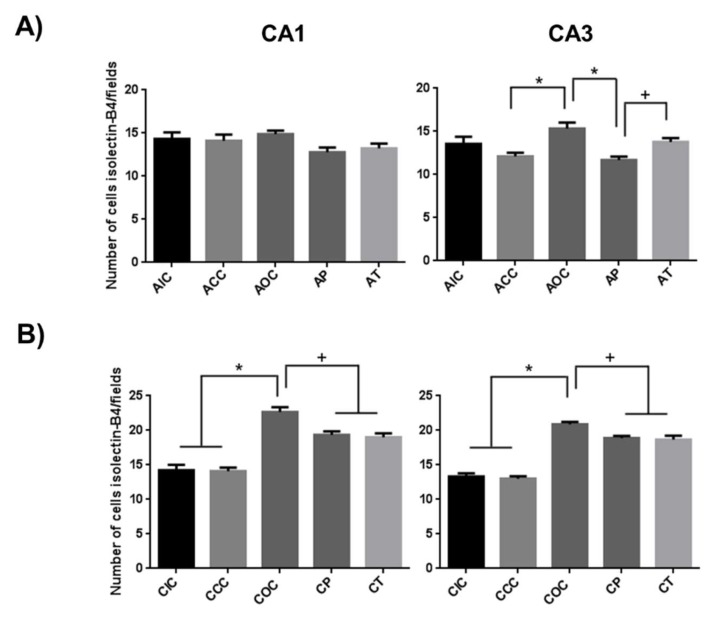
Effect of CUR is depicted on the total count of microglial cells in the acute and chronic exposure to O_3_. (**A**) The total number of isolectin-B4 positive cells in the acute phase is shown in the graph as mean ± SEM. In both CA1 and CA3 regions, no statistical differences were observed after 15 days of exposure. Meanwhile, the total microglial cell count was reduced in the AP group due to CUR. (**B**) The total isolectin-B4 positive microglial cells in the chronic phase are shown. The cell count increased in the group exposed to O_3_. In contrast, CUR decreased the total number of microglial cells in CP and CT groups in both hippocampal regions. * *p* < 0.001, + *p* < 0.01.

**Figure 6 molecules-24-02839-f006:**
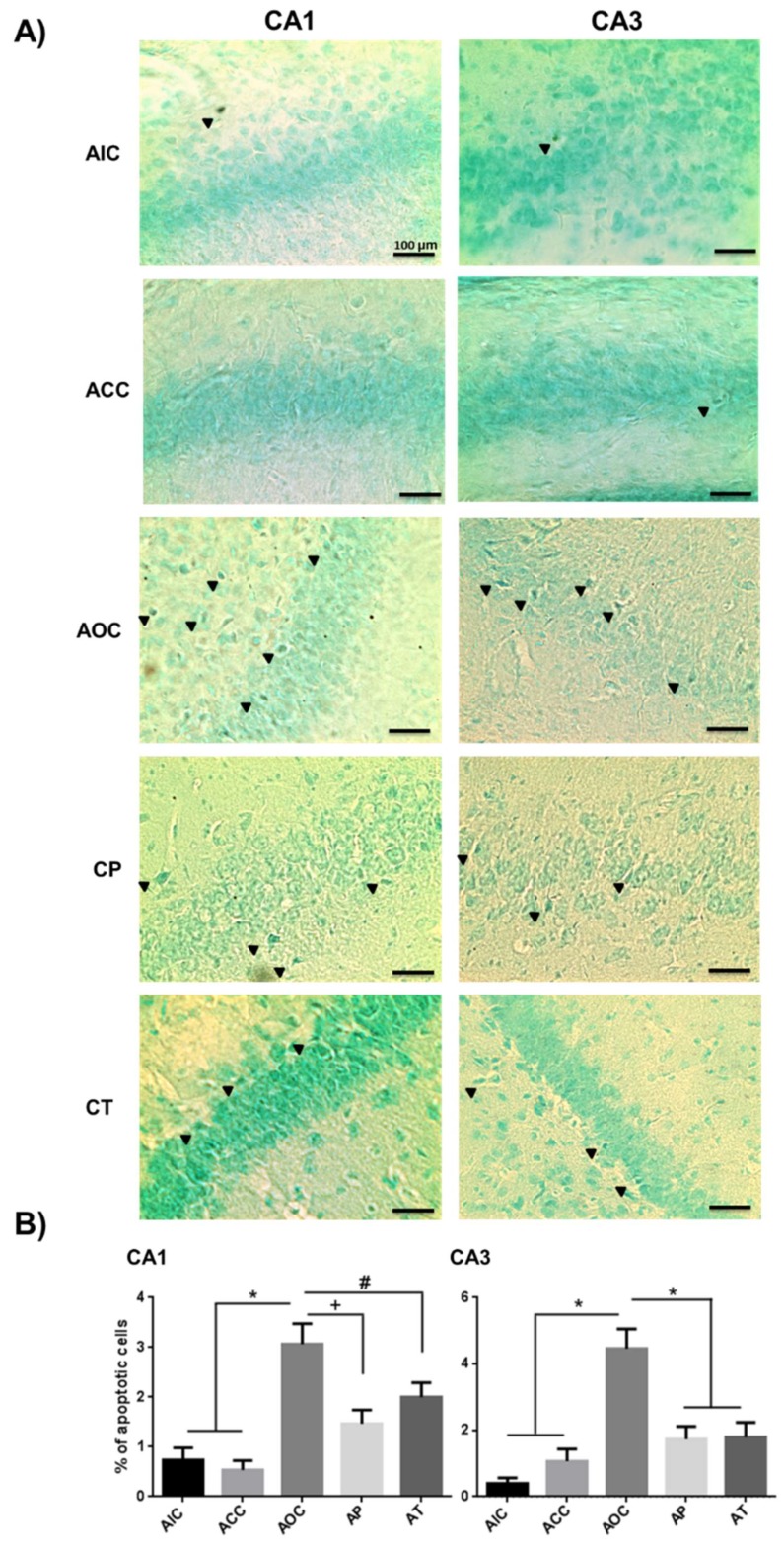
Effect of CUR on the quantitation of apoptotic hippocampal cells in the acute phase. (**A**) Representative images of apoptotic cells (black arrows) in the CA1 and CA3 regions of the rat hippocampus in the acute phase are shown. The AOC group developed a higher percentage of apoptotic cells than those found in the AP and AT groups. Scale bars 100 μm. (**B**) Total percentage of apoptotic cells are shown in the graph as mean ± SEM. * *p* < 0.001, + *p* < 0.01, # *p* < 0.05.

**Figure 7 molecules-24-02839-f007:**
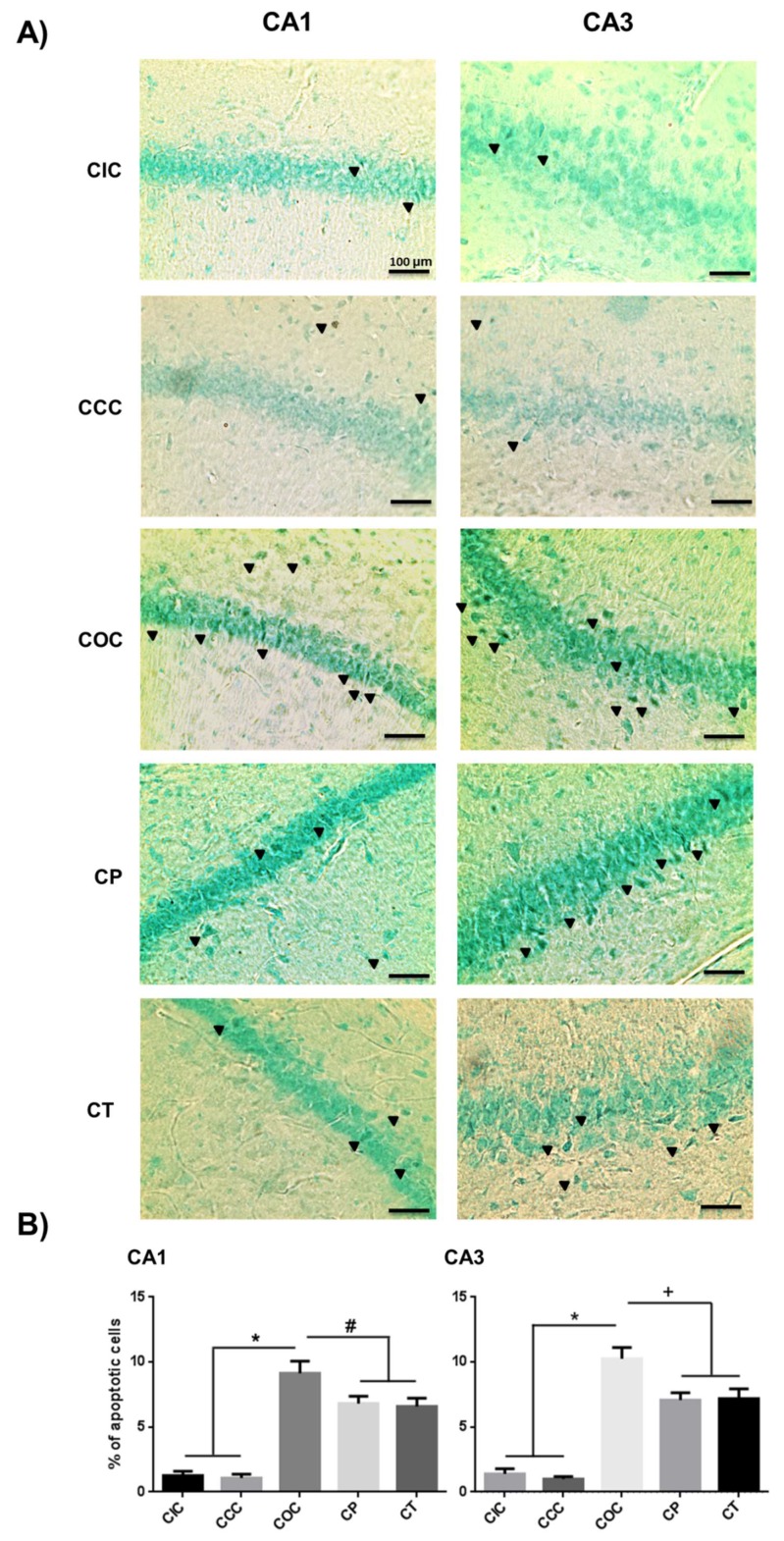
Effect of CUR on the quantitation of apoptotic hippocampal cells in the chronic phase. (**A**) Representative images of apoptotic cells (black arrows) in the CA1 and CA3 regions of rat hippocampus in the chronic phase are shown. The CP and CT groups presented a significantly decreased percentage of apoptotic cells than the COC group. Scale bars 100 μm. (**B**) Total percentage of apoptotic cells are shown as mean ± SEM. SEM. * *p* < 0.001, + *p* < 0.01, # *p* < 0.05.

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
