# Peer review of "Dietary Curcumin Prevented Astrocytosis, Microgliosis, and Apoptosis Caused by Acute and Chronic Exposure to Ozone"

_molecules, 2019, doi:10.3390/molecules24152839_

Round 1
Reviewer 1 Report
The manuscript "Dietary curcumin prevented astrocytosis, microgliosis and apoptosis caused by acute and chronic exposure to ozone" needs revision, as follows:
- the full name of each abbreviation should be given when using it at the first time, such as "NF-kB" (Abstract);
- please, change the following sentence "There is scientific interest in evaluating" to "There is interest", due to a question of style;
- Introduction, second paragraph: both acute and chronic inflammation may occur in the CNS due to redox impairment. Therefore, it is suggested removing "chronic" from that sentence;
- Introduction, second paragraph: please, it is needed indicating, in a summarized form, why the hippocampus is more sensitive to redox impairment when compared to other brain areas. Also, please cite the brain areas that are being compared with the hippocampus. The substantia nigra, for example, may be more sensitive to redox impairment than the hippocampus due to the high levels of iron, copper, dopamine, dopaminochrome, among others. This part of the Introduction needs to be revised and a new and clearer text should be written;
- Introduction, third paragraph: suggestion - "Astrocytes are microglial cells...";
- Introduction, third paragraph: please, avoid using "they" (such as in "They modulate"). Suggestion: Those cells modulate... OR Astrocytes modulate... OR something similar;
- the Introduction contains very short paragraphs. It should be avoided;
- the Introduction needs additional information regarding the effects of curcumin on the mitochondria. Alteration in mitochondrial dynamics, as may be induced by curcumin, affects mitochondria-related redox environment and bioenergetics. Therefore, it is suggested reading the following reference in order to better explain why the Authors have investigated curcumin in the field of redox environment (Biotechnol Adv. 2016 Sep-Oct;34(5):813-826. doi: 10.1016/j.biotechadv.2016.04.004);
- the figures of the manuscript are of excellent quality;
- Results section: it is not necessary citing the values obtained during calculation, such as "CA1, 55.22 ± 1.48; CA3 49.56 ± 1.37". It is suggested just adding the "p" value;
- the data obtained by the Authors have been done in a succint form and is not speculative;
- the figures related with fluorescent photomicrographs are excellent and a suggestion may be made to future research: studying the effects of curcumin in neuroblast proliferation and neuronal differentiation in animals subjected to this same experimental model (neurogenesis studies);
- Discussion section, 5th paragraph: "The preventive and therapeutic administration of CUR was" should be "were", since the Authors are referring to both preventive and therapeutic ways of administration (plural);
- the Discussion section also presents very short paragraphs, and it is suggested altering it due to a question of style;
- the utilization of abbreviations needs to be corrected throughout the manuscript, as commented above.
Author Response
AUTHOR NOTES TO REVIEWER 1:
According to your kind questions, we are answering about the changes made to the manuscript titled: Dietary curcumin prevented astrocytosis, microgliosis and apoptosis caused by acute and chronic exposure to ozone.
1. The abbreviations in abstract as well in the entire new document have been defined: Nuclear Factor-kappa B (NF-κB).
2. The syntax corrections along the document were made in the new version as kindly suggested.
3. In the Introduction section in the second paragraph, the word "chronic" has been removed.
4. The information about the sensitivity of hippocampus to oxidative damage caused by ozone compared to other CNS regions is now described in the second paragraph (Lines 50 to 58).
5. In the third paragraph, the expression – “Astrocytes are microglial cells” is not possible, because astrocytes are different from microglial cells. The sentence has been corrected in order to clarify the idea.
6. In the same paragraph (line 60-63), the word “They” has been omitted and the sentence has been re-written.
7. The introduction now does not contain short paragraphs.
8. The information about the mitochondrial function relevance in the maintenance of the redox state in presence of curcumin is now expressed in the seventh paragraph (lines 94 to 97).
9. We appreciate the comments about the quality of the figures.
10. In the results section, we have omitted the absolute data and the “p” value was maintained.
11. We thank the comment about the quality and reliability of our results.
12. We appreciate the suggestion to study the neurogenesis in rats exposed to ozone in presence and absence of the influence of curcumin. This will be an important project to be performed in our Lab.
13. In the discussion section, the verb has been corrected. We apologize for that, but apparently the software made the change.
14. The abbreviations have been corrected throughout the manuscript.
Reviewer 2 Report
The work of Nery-Flores et al, on the protective effect that curcumin has on the central nervous system from the damage caused by ozone, is very interesting, the methodology well designed.
I suggest expanding the manuscript on the following:
His work conclusion wasIn conclusion:
the results of this work demonstrate that the administration of CUR decreases the astrocytosis, microgliosis, and apoptosis in the hippocampus induced by exposure to O3. CUR could be a safe and efficient alternative to prevent CNS damage caused by exposure to O3 and other pollutants.
What is the mechanism of action they propose of curcumin?
What is the participation of the NRF2 transcriptional factor and curcumin?
Can ozone activate NRF2?
What other phytochemical compounds have been reported to protect from ozone damage? And what mechanism is postulated? Can you compare the different phytochemicals with curcumin?
Author Response
AUTHOR NOTES TO REVIEWER 2
According to your kind questions, we are answering about the changes made to the manuscript titled: Dietary curcumin prevented astrocytosis, microgliosis and apoptosis caused by acute and chronic exposure to ozone.
1. We would like to thank the comments to our work; we consider that the study of natural molecules like curcumin, resveratrol, epigallocatechin, capsaicin, among others is important to offer potent and safe protection against the deleterious effects of pollutants in air, water or soil. We hope that our results could contribute to the proposal of new studies in human beings in order to validate the potential of these substances to be used extensively in diverse human populations.
2. The conclusion section has been corrected.
3. The mechanisms of action of curcumin include its action as ROS scavenger, suppressor of NFκB activation, activator of Nrf2, and suppressor of p53. However, dynamics of such mechanisms show variations depending on the time when curcumin is administrated. This means that the therapeutic administration could be focused primary to reduce the previous activation of NFκB which leads to a decreased expression of inflammatory cytokines and other genes under control of NFκB. In a secondary action, curcumin may increase the expression of anti-oxidant enzymes by activating Nrf2 through modification of the sulfhydryl groups of Keap1. In a preventive mode, curcumin may first activate Nrf2 and increasing the expression of anti-oxidant enzymes that act simultaneously with curcumin to neutralize ROS. These actions contribute to avoid an intense activation of NFκB through inhibition of IKK phosphorylation when the oxidative aggression encounters an environment reinforced at different points. However, in chronic exposure the oxidant aggression overcomes this reinforcement and damage may occur. This could be improves increasing the curcumin dose.
4. Ozone may act as an activator for Nrf2 at low concentration (under 0.3 ppm) and for short-term exposure (Cho et al., 2013). However, at higher ozone concentration and larger exposure periods the antioxidant response elicited by ozone is overcome and oxidative damage increases (Nery-Flores et al., 2018)
5. The only natural molecules that have been studied against other air pollutants than ozone are vitamin C and E, but such studies do not analyze the damage to the CNS, however the mechanisms of action are mainly as ROS scavengers. Synthetic molecules like tibolone (Pinto-Almazán et al., 2014) and taurine (Rivas-Arancibia et al., 2000) have been previously reported against damage caused by ozone in CNS, but the risk of side effects limits the administration these molecules. The protective effect of capsaicin against inflammation of airways of Guinea pigs caused by ozone has been reported (Kaneko et al., 1994)
Cho H-Y, Gladwell W, Yamamoto M, Kleeberger SR (2013) Exacerbated Airway Toxicity of Environmental Oxidant Ozone in Mice Deficient in Nrf2. Oxidative Medicine and Cellular Longevity 2013:14.
Kaneko T, Ikeda H, Fu L, Nishiyama H, Matsuoka M, Yamakawa HO, Okubo T (1994) Capsaicin reduces ozone-induced airway inflammation in guinea pigs. American Journal of Respiratory and Critical Care Medicine 150:724-728.
Nery-Flores SD, Mendoza-Magana ML, Ramirez-Herrera MA, Ramirez-Vazquez JJ, Romero-Prado MMJ, Cortez-Alvarez CR, Ramirez-Mendoza AA (2018) Curcumin Exerted Neuroprotection against Ozone-Induced Oxidative Damage and Decreased NF-kappaB Activation in Rat Hippocampus and Serum Levels of Inflammatory Cytokines. Oxid Med Cell Longev 2018:9620684.
Pinto-Almazán R, Rivas-Arancibia S, Farfán-García ED, Rodríguez-Martínez E, Guerra-Araiza C (2014) Efecto neuroprotector de la tibolona contra el estrés oxidativo inducido por la exposición a ozono. RevNeurol 58:0441-0449.
Rivas-Arancibia S, Dorado-Martı́nez C, Borgonio-Pérez G, Hiriart-Urdanivia M, Verdugo-Dı́az L, Durán-Vázquez A, Colin-Baranque L, Rosa Avila-Costa M (2000) Effects of Taurine on Ozone-Induced Memory Deficits and Lipid Peroxidation Levels in Brains of Young, Mature, and Old Rats. Environmental Research 82:7-17.